# Biotechnological Applications of Mushrooms under the Water-Energy-Food Nexus: Crucial Aspects and Prospects from Farm to Pharmacy

**DOI:** 10.3390/foods12142671

**Published:** 2023-07-11

**Authors:** Xhensila Llanaj, Gréta Törős, Péter Hajdú, Neama Abdalla, Hassan El-Ramady, Attila Kiss, Svein Ø. Solberg, József Prokisch

**Affiliations:** 1Institute of Animal Science, Biotechnology and Nature Conservation, Faculty of Agricultural and Food Sciences and Environmental Management, University of Debrecen, 138 Böszörményi Street, 4032 Debrecen, Hungary; 2Plant Biotechnology Department, Biotechnology Research Institute, National Research Centre, 33 El Buhouth St., Dokki, Giza 12622, Egypt; 3Soil and Water Department, Faculty of Agriculture, Kafrelsheikh University, Kafr El-Sheikh 33516, Egypt; 4Knowledge Utilization Center of Agri-Food Industry, University of Debrecen, Böszörményi út 138, 4032 Debrecen, Hungary; 5Faculty of Applied Ecology, Agriculture and Biotechnology, Inland Norway University of Applied Sciences, 2401 Elverum, Norway

**Keywords:** mushroom farming, nanoremediation, bioenergy, nanobiotechnology, agro-wastes, nanoparticles

## Abstract

Mushrooms have always been an important source of food, with high nutritional value and medicinal attributes. With the use of biotechnological applications, mushrooms have gained further attention as a source of healthy food and bioenergy. This review presents different biotechnological applications and explores how these can support global food, energy, and water security. It highlights mushroom’s relevance to meet the sustainable development goals of the UN. This review also discusses mushroom farming and its requirements. The biotechnology review includes sections on how to use mushrooms in producing nanoparticles, bioenergy, and bioactive compounds, as well as how to use mushrooms in bioremediation. The different applications are discussed under the water, energy, and food (WEF) nexus. As far as we know, this is the first report on mushroom biotechnology and its relationships to the WEF nexus. Finally, the review valorizes mushroom biotechnology and suggests different possibilities for mushroom farming integration.

## 1. Introduction

Water, energy, and food (WEF) are all important for human survival. A considerable volume of studies in the literature are attributed to the global water, food, and energy nexus (e.g., [1,2,3]). These studies mainly focused on the scarcity of these resources [4,5], their potential impact on human health [1], their nexus security [6], assessment of WEF using geographic information systems (GIS) [7], optimizing the WEF nexus under different agricultural systems such as the agro-forestry–livestock system [8], and the risks to this nexus [9]. The WEF nexus is essential for socioeconomic and sustainable development. Many global changes (e.g., sea level rise, higher temperatures, and extreme weather events) may influence WEF and its scarcity risk [9]. Furthermore, global stresses, such as the recent COVID-19 pandemic and the ongoing Russia–Ukraine war, have worsened global energy and food security problems [10]. These challenges force us to search for alternatives to WEF resources and to rethink our approaches to minimize the loss of resources [10]. It has been reported that 4 trillion megajoules of energy and 82 billion cubic meters of water are lost globally every year. Furthermore, 344 million tons of global avoidable food waste is responsible for squandering 4 × 10^18^ joules of energy and 82 billion m^3^ of water [11].

A great deal of previous research into mushrooms has focused on applications in the agricultural, industrial, and medicinal sectors. A mushroom is the fruit body of a fungus. There are 44,000 known species in the fungi kingdom, but not all of them produce mushrooms. Fungi lack chlorophyll and are heterotrophic organisms that break down organic matter in various ways [12]. The applications of mushrooms in food and medicine have been widely investigated (e.g., [13,14,15]). More recent attention has focused on the global issues of water, energy, and food from different points of view, including mushrooms as bioresources for nutraceuticals and food [16], therapeutic values [17], producing biofuel [14], producing high-protein food [18], and water remediation using mushroom residue [19,20]. Collectively, these studies outline a critical role for mushrooms in our life and support the hypothesis that mushrooms have a crucial role to play in different fields. An important theme emerging from the discussion of these studies is the water–energy–food nexus.

Recently, several studies have been published on a variety of mushroom species and their relationships with health benefits. Examples include the shiitake mushroom (*Lentinula edodes*) [17]; the biotechnological applications of the *Yarrowia lipolytica* mushrooms [21]; the green biotechnology of *Pleurotus ostreatus* L. [22]; and the biorefinery abilities of *Pleurotus ostreatus* mushrooms [23]. A strong relationship between mushroom species and their biotechnological attributes has been reported in the literature. Various studies have provided insights into various fields, including the production of enzymes such as cellulase [20], sugar alcohols such as arabitol, erythritol, and mannitol [21], and the processes that lead to the production of biofuels or the use of bioenergy [24,25,26]. Hence, it could conceivably be hypothesized that mushrooms have a crucial role to play in the production of food and energy and that there is a strong relationship between mushrooms and the water–energy–food nexus.

## 2. Methodology of the Review

The current study was designated to highlight the role of mushrooms in the production of food and energy with a focus on their biotechnological applications (Figure 1). After the formulation of the goal as described above, keywords were identified and database searches were conducted (mainly PubMed, ScienceDirect, SpringerLink, Frontiers, and MDPI). The keywords included were “water–food–energy nexus”, “edible mushrooms”, “medicinal mushrooms”, “toxic mushrooms”, “mushroom farming”, “mushrooms and space”, “mushroom nanobiotechnology”, “mushrooms and nanoparticles”, “mushrooms and nanoremediation”, “mushrooms and bioenergy”, “mushrooms bioactives”, “mushrooms and medicine”, “mushrooms biorefinery”, etc. The article search was refined to cover only the last five years. We focused on articles from journals with high impact factors and/or a good reputation. After compiling the articles, they were sorted according to the tentative sections planned for this review.

## 3. Water-Energy-Food Nexus

Water and energy are crucial to food production. Thus, intensive scientific research on water, energy and food is needed to reduce poverty for human wellbeing and sustainable development (Figure 2). It is possible that natural resources (water, energy, and food sources) benefit (i.e., there are positive outcomes) from various practices, including food manufacturing, desalination, or the valorization of municipal waste, but negative results may appear due to climate change, over-exploitation, emissions, and pollution. Paying close attention to the WEF nexus is essential to support food security, sustainable agriculture, and overall human and environmental health. Therefore, there is an urgent need to manage global water, energy, and food resources by supporting their sustainable use, particularly when there is a scarcity of these resources [27,28]. The WEF nexus is linked to many global issues, for example, climate change [5], the sustainable use of resources [2], urban ecosystem services [29], sustainable irrigation systems [30], global water resources allocation [31], optimizing different agro-ecosystems like the agro-forestry–livestock system [8], managing risk levels of available WEF resources [32], the co-production of WEF resources [33], urban green and blue infrastructure [34], and predicting WEF security in the future [6].

Although the WEF nexus is a global issue, many studies have been focused on the local level. These include case studies in Mexico [6], Egypt [7], China [8,30,35,36], Romania [37], Iran [38], and in Africa [33], to name a few. Similar global stressors are faced in regions all over the world. There is a need to find ways to supply proper and adequate water, energy, and food under globalization, urbanization, economic growth, and climate change [39]. Humanity faces serious problems in supplying enough water, energy, and food to meet global needs. These problems will be aggravated in the future. Thus, to ensure WEF security and sustainability, it is crucial to understand the factors and dynamics that drive WEF production and co-production to develop policies and plans more effectively [33].

## 4. Mushroom Farming: Basics and Requirements

Mushrooms are being referred to as the new superfood due to their health benefits. It is thought that the benefits of mushrooms were already well-known 400 million years ago, discovered by Egyptian and Roman civilizations as well as by the ancient Greeks. The latter called mushrooms the “*Food of the Gods*” [16]. It is not possible to cultivate all mushroom species, and a special protocol should be followed for any cultivation. The protocols should include certain specific steps and requirements for production (Figure 3). The existing mushroom farming systems are presented in Figure 4. The main aim of mushroom production is food, but different farming systems can co-exist with different aims, such as a mushroom–forest system, mushrooms for livestock feeding, mushrooms for bee breeding, urban mushroom farming, and smart mushroom farming. Other major applications for mushrooms include their use in the remediation of polluted soil and water as well as in the production of enzymes, bioactives, and bioenergy.

The value of mushrooms as food has increased after the COVID-19 pandemic and the global food crisis that followed. This impact has been recorded all over the world in many zones and several countries (e.g., [40,41]). This global issue led many countries to search for smart and novel food processing technologies [42] and healthy food alternatives like mushrooms [16]. The role of mushrooms with bacteriocin nanocomposite in fighting the COVID-19 pandemic was confirmed by inhibiting metal NPs-bacteriocin [43]. Mushrooms promote human immunity, as reported during the COVID-19 pandemic [16]. More light will be shed on this, and we will discuss in detail the applications of mushrooms in the following sections.

Mushrooms have low water and energy requirements for cultivation. In general, the main factors that control mushroom cultivation include relative humidity, aeration, temperature, and contamination [44]. Adding a small amount of water to cultivated mushrooms every day can provide the required humidity [44]. The total primary energy consumed by *Agaricus bisporus* is 29.1 MJ (27.8 and 1.3 megajoule per kilogram from non-renewable and renewable sources, respectively) per kilogram of mushroom product [45]. Producing 1.0 kg of mushrooms can generate 0.71 kg of CO_2_, whereas the amount for field-grown vegetables was 0.37 kg CO_2_-eq/kg [46]. In general, each 1.0 kg of mushroom product can generate about 5.0 kg of spent mushroom substrate (SMS) [47], whereas only 2.5 kg of fresh SMS results from the production of 1 kg of fresh *A. bisporus* mushrooms [48].

## 5. Mushrooms as a Healthy Food

Due to their high nutritional value and functional potential, mushrooms are known as ideal food supplements, a new super-food, next-generation food of the future, and myco-protein food. There is a strong relationship between mushrooms and the WEF nexus (Figure 5). The relationship has positive as well as negative aspects. Mushrooms have low requirements of water and energy for cultivation, they produce bioenergy and obtain clean water via myco-remediation, are vital for biorefineries, and make ideal food supplements. The observed negative correlations between mushroom and the WEF nexus include air and water pollution caused by spores, and the fact that some mushroom species are considered poisonous, causing health problems.

Edible mushrooms are rich in proteins (30–48%), ash (7–17%), dietary fiber (16–20%), carbohydrates (12.5–40%), minerals, and vitamins like B, C, D, and E. Mushrooms also have low calorie, cholesterol, fat, and sodium contents [15,16,49,50]. Mushrooms contain a variety of bioactives including alkaloids, lactones, polyphenolic compounds, polysaccharides, sesquiterpenes, sterols, and terpenoids [51]. Mushrooms may have more than 100 bioactive extracts and 156 compounds such as eminent antibiotics [52]. Mushrooms are rich in B vitamins, including thiamine (B_1_), riboflavin (B_2_), niacin (B_3_), pantothenic acid (B_5_), and folate (B_9_), which produce red blood cells that carry oxygen throughout the human body [49].

Recently, several reports have been published on the nutraceutical and therapeutic values of different species of edible mushroom [15,51,53,54], such as porcini (*Boletus edulis*) [55] and shiitake (*Lentinula edodes*) [17] mushrooms, which have health-promoting effects (Table 1). These reports emphasized that mushrooms have bioactive compounds related to proteins and peptides as fungal immunomodulatory proteins, including ubiquitin-like proteins, lectins, and some enzymes such as ergothioneine, laccases, and ribonucleases [56]. The common benefits of mushrooms may include their antibacterial, antifungal, antioxidant, antiviral, antihypertensive, immunomodulatory, antitumor, antihypercholesterolemic, antihyperlipidemic, antidiabetic, and anti-inflammatory properties [56,57].

## 6. Mushroom Biotechnology

Mushroom biotechnology is defined as the science in which mushrooms are included in processes like bioconversion, biorefining, bioremediation, and biodegradation (Figure 6). Negative and positive issues can arise from the relationship between mushrooms and WEF (Figure 6). Biotechnology might guarantee the quality and security of WEF and thus contribute to combatting infectious diseases, reducing hunger, and remediating environmental degradation. The main problems within this relationship are represented by the production of biological weapons and security/safety problems at the global level of WEF.

The many biotechnological applications of mushrooms have generated great attention aimed at the growing demand for energy, food, fodder, and fertilizers [67,68,69]. Biotechnological applications of mushrooms are considered an emerging approach for utilizing WEF resources. In this section, four subsections will be presented, including the use of mushrooms to produce nanoparticles, bioactive compounds, bioenergy, and for nanoremediation.

### 6.1. Mushrooms to Produce Nanoparticles

Myco- or green biosynthesis of eco-friendly nanoparticles (NPs) is one of the most important recent applications of mushrooms. This biological method for producing NPs is preferable compared to chemical or physical methods to avoid environmental pollution. NPs have great applications in the fields of water, energy, and food, including enormous benefits in water purification/remediation [70], producing high-efficiency energy and its storage [71] and food security [72]. In this subsection, we will focus on different applications of NPs in the fields of food and energy, whereas water will be discussed in the next subsection. Nanoparticles produced by mushrooms have several applications in the food and human health sectors, as tabulated in Table 2. Silver nanoparticles are the most common among NPs, which can be produced using the mushroom *Laxitextum bicolor* for myco-synthesis, as reported by [73].

Mushrooms can produce NPs through two different methods: (1) production inside the mycelium cells stimulated by intracellular enzymes or (2) production outside these cells by extracellular enzymes [74,75,76]. The myco-synthesis of NPs can be performed by certain steps, as presented in Figure 7. Myco-synthesis can produce gold (Au-NPs), silver (Ag-NPs), selenium (Se-NPs), magnesium oxide (MgO-NPs), titanium oxide (TiO_2_-NPs), copper oxide (CuO-NPs), zinc oxide (ZnO-NPs), and cadmium sulfide (CdS-NPs) NPs [77]. Silver NPs are very common and can be generated using many mushroom species (*Agaricus bisporus*, *Amanita muscaria*, *Pleurotus ostreatus*, *Ganoderma applanatum*, etc.). Ag-NPs have been utilized for their antibacterial, anticancer, antioxidant, and antimicrobial activities [77].

**Table 2 foods-12-02671-t002:** Selected studies on the biosynthesis of nanoparticles (NPs) by mushrooms and their suggested applications related to water, energy, food, and human health.

Mushroom Species	Nanoparticles (NPs)	The Application	References
Enoki mushroom (*Flammulina velutipes*)	Ag-NPs (~10 nm)	Biodegradable natural biopolymers as active food packaging films.	[78]
Oyster mushroom (*Pleurotus florida*)	Ag-NPs (~12.62 nm)	Effective antimicrobial agents as an alternative to traditional antibiotics to control diseases/microbial infection.	[79]
Reishi mushroom (*Ganoderma lucidum*)	Ag-NPs (15–22 nm)	Ag-NPs can be applied in the pharmaceutical, medical, and cosmetic fields due to their antioxidant, antibacterial, and antifungal activity.	[80]
Oyster mushroom (*Pleurotus florida*)	Gold-platinum (Au-Pt-NPs, 16 nm)	Au-Pt-NPs showed anticancer activity against human colon cancer.	[81]
*Pleurotus giganteus*	Ag-NPs (2–20 nm)	Ag-NPs have antibacterial and α-amylase inhibitory activity.	[82]
*Macrolepiota procera*	Ag-NPs (20–50 nm)	Ag-NPs have antibacterial activity as a green corrosive inhibitor for mild steel in cooling tower water systems.	[83]
*Termitomyces heimii* mushroom	CdS-NPs (<5 nm)	CdS-NPs have potential use in energy (solar panels), biomedical, biofilm, drug delivery, and environmental applications.	[84]
*Cordyceps militaris* mushroom	ZnO-NPs (1.83 nm)	ZnO-NPs can be used for the development of therapeutic drugs and have antioxidant, antidiabetic, and antibacterial activity.	[85]
*Inonotus hispidus* mushroom	Ag-NPs (69.24 nm)	Ag-NPs exhibited activity against different pathogenic bacteria and fungi, showing antimicrobial potential.	[86]
*Ramaria botrytis* mushroom	Ag-Au bimetallic composite NPs	The nano-composite was effective for intensive industrial and biomedical applications due to powerful antioxidant properties for DPPH radical scavenging.	[87]
Shiitake mushroom (*Lentinula edodes*)	ZnO-NPs (21–25 nm)	ZnO-NPs degraded methylene blue dye pollution by 90% within 135 min in wastewater. It also showed promise as an antibacterial product.	[88]
Portabello mushroom *(A. bisporus)*	Au-NPs (53 nm)	Au-NPs reduced methylene blue by about 98% in wastewater and decolorized azo dye.	[89]
*Ganoderma lucidum*	ZnO-NPs (using 25 mL for extraction)	ZnO-NPs were used in vitro as nanofertilizer for feeding garden cress (*Lepidium sativum*).	[90]
Edible mushroom (*A. bisporus*)	Ag-NPs (average 17 nm)	Myco-fabricated Ag-NP had antioxidant/antimicrobial effects without any cytotoxic impacts on human dermal fibroblast cells.	[91]

A proposed mechanism for forming NPs from mushrooms has been discussed in many previous publications using different types of enzymes (mainly reductases), intracellular or extracellular, for reducing and stabilizing NPs through the mushroom exudates of biomolecules [92,93,94]. Mushroom hypha cells can secrete exudates as extracellular enzymes and reduce the oxidation state of metal ions, creating elemental forms of these metals through secondary metabolites such as alkaloids, cyclosporine, griseofulvin, flavonoids, lovastatin, polysaccharides, mevastatin, and saponins [77]. The intracellular enzymes work in the mushroom hypha cells in the presence of enzymes such ACCases (Acetyl-CoA carboxylase), nicotinamide adenine dinucleotide (NADH), NADH-dependent nitrate reductase enzymes, and peroxidases, which can reduce metallic ions into reduced metallic elemental forms (M^0^), creating NPs. In both intra- and extra-cellular methods, NPs should be purified to eliminate any remaining fungal by-products through simple filtration and then centrifuging or chemical washing [75].

### 6.2. Mushrooms for Bioremediation

Soil and water pollution resulting from rapid industrialization, the intensive use of agrochemicals, including mineral fertilizers and pesticides, urbanization, and other anthropogenic activities is a serious global problem. Using plants to remove pollutants or combining plants and microbes in phytoremediation has been known for the last several decades, whereas myco-remediation has gained recent attention. Such remediation depends on certain enzymes that the mushrooms can produce and that can be used in the degradation of organic pollutants [95]. Myco-remediation can also be accomplished by applying spent mushroom substrate (SMS) as a by-product after mushroom cultivation. This has many advantages, including its eco-friendly nature and low cost (Table 3) [96]. The potential of myco-remediation can be increased by integration with nanomaterials, leading to the nano-restoration of polluted soil and water [97,98]. Many recent studies confirmed that mushroom remediation is a sustainable and promising approach for the biodegradation of persistent pollutants in soil and wastewater treatments through the production of enzymes such as peroxidase and laccase (e.g., [95,98]).

### 6.3. Mushrooms to Produce Bioenergy

Due to its nutritional value and functional bioactivities, the direct product of mushroom cultivation is healthy food. This builds global market value and has led to steady growth in the mushroom industry [14]. Due to the ability of recycling and utilizing mushroom residues, the cultivation of mushroom is considered an excellent biotechnological process (Figure 8). After harvesting mushrooms, a huge amount of waste (spent mushroom substrate; SMS) remains. It is urgent that these wastes be managed in a sustainable way that protects the environment. Based on the concept of waste-to-fuel, one management option for SMS lignocellulosic wastes is to use it as a feedstock to produce biofuels, including bioethanol, biogas, bio-H_2_, bio-oil, and solid biofuels [14]. Therefore, integrated mushroom cultivation for food and biofuel production can simultaneously meet rapidly rising global demands for both energy and food [14]. The cultivation of mushrooms can serve as an efficient biological pretreatment for producing biofuel and promoting its yield, improving the overall economy and supporting the biorefinery approach [108].

There are increasing concerns within the mushroom industry about the accumulation of SMS. If SMS waste is not properly managed, this may have an adverse impact on the environment, economy, and human health. Therefore, there is an urgent need for an effective strategy for the proper management of SMS by recycling and reutilization. Several studies focused on this, and six examples are provided here: (1) Study the valorization of SMS for producing low-carbon biofuel viewed through the circular economy of utilizing and recycling SMS as renewable feedstock to produce biogas, biohydrogen, bioethanol, bio-oil, and solid biofuels [14]. (2) Assess optimal conditions to increase the yield of biogas from SMS using the hydrothermal pretreatment (HTP) method to improve the biodegradability of the SMS by 87% compared to mechanically pretreated biodegradability of 61% [109]. (3) Combine SMS with sewage sludge (SS) to convert SS into renewable fuels and N-rich liquid fertilizers through hydrothermal carbonization while also significantly improving fuel and fertilizer quality [24]. (4) Apart from producing bioenergy, SMS can be used to produce compost through the enzyme activity of polyphenol oxidase, carboxymethyl cellulase, catalase, and laccase, with are correlated with the composition of the microbial community [110]. (5) Applying liquid organic fertilizer formed from anaerobic fermentation of liquid SMS enhanced pak choi production by around 30% and improved the level of nutrients in the studied soil due to the synthesized hormone indole-3-acetic acid, IAA [111]. (6) SMS extract efficiently protected the active components of *Bacillus thuringiensis* (Bt) from UV irradiation by forming lignin and lignin–carbohydrate complexes, which possessed the ability to scavenge reactive oxygen species (ROS), had a high UV-screening effect, and improved the UV stability of Bt formulation [112].

### 6.4. Mushrooms to Produce Bioactives

Mushrooms are rich in pharmaceutical and nutritional compounds. These bioactives have a variety of clinical applications and many therapeutic attributes because of their qualities as antioxidants, as well as anticancer, antimicrobial, antidiabetic, anti-inflammatory, and prebiotic activities [113]. Several recent studies on mushroom bioactives discussed the compounds found in different groups of mushrooms (Table 4) and confirmed the benefits of edible/medicinal mushroom as a source of healthy food [49]. Kour et al. [16] reported on the nutraceuticals found in mushrooms and their benefits as food due to their content of bioactives, such as polysaccharides protein complexes, polysaccharides, peptides, terpenoids, and phenolic compounds. The immunomodulatory effect of mushrooms as anticancer foods was confirmed due to the existence of many phytoconstituents (e.g., lentinan, maitake-D fraction, and schizophyllan), which can upregulate the production of cytokine, cause cell cycle arrest, and mediate cytotoxicity [114]. The bioactives in edible mushroom spores and their quality as a novel resource for both food and medical compounds were reported by Li et al. [115]. These bioactive compounds may include the following groups: polysaccharides, amino acids, alkaloids, fatty acids, nucleosides, triterpenes, and others. The role of bioactive metabolites in edible mushrooms in preventing human hair loss was investigated by Tiwari et al. [116]. They reported that hair loss could be due to the existence of androgenic alopecia (AGA), which occurs because of the hyperactivity of the steroid 5α-reductase2. Many review articles have been published on mushroom bioactives and their therapeutic potential in general (e.g., [15,16,117,118,119,120]), or with focus on certain bioactive compounds such as proteins [121,122], polysaccharides [123,124,125], non-peptide secondar + y metabolites [126], terpenoids [127,128], etc.

## 7. Mushrooms: Pros and Cons

Several benefits of mushrooms have been reported, mainly in the fields of agriculture, medicine, pharmacology, and the food industry. The most crucial benefits of mushrooms can help achieve many UN development goals (SDGs) (Figure 9). These goals target water, energy, and food security at a global level for a bio-based circular economy [68]. It is worth noting the crucial role mushrooms play in supporting the WEF nexus under the SDGs. Myco-biotechnology can improve the myco-cell factories, which can help meet 11 out of 17 SDGs [68].

Mushrooms may cause allergic reactions affecting the skin, nose, throat, and lungs, with additional problems arising from toxic or poisonous mushrooms. Poisonous mushrooms have toxins that pose a threat to human health and safety. Poisonous mushrooms can be divided into six groups based on the symptoms they cause (Table 5), including cytotoxic, myotoxic (rhabdomyolysis), neurotoxic, gastrointestinal irritation, metabolic (including endocrine and related toxicity), and miscellaneous adverse reactions [137]. Although the annual global number of fatalities resulting from poisonous mushrooms is unknown, cytotoxic and myotoxic poisoning from mushrooms are the most lethal poisonings and can cause death [137]. The most common species of poisonous mushrooms include *Agaricales*, *Pezizales* (Ascomycota), *Russulales* (Basidiomycota), and *Boletales* [138]. Poisonous mushrooms have a negative impact on human health, but many benefits can be achieved using the toxins of such mushrooms as tools for research on developmental biology, structural biology, and cell biology. Therefore, more research on poisonous mushrooms is needed to explore possible beneficial applications of such mushrooms [137].

Studies of poisonous mushrooms have reported on a wide variety of topics, including using poisonous mushrooms to extract non-peptide secondary metabolites for the development of drugs [126], the potential global benefits and problems of poisonous mushrooms [137], the investigation of amino-group-containing mushroom toxins (i.e., muscimol, ibotenic acid, 2-amino-4-pentynoic acid, and 2-amino-4,5-hexadienoic acid) that can cause hallucination and neurotoxicity in humans [144], ustalic acid in *Tricholoma ustale* as a toxin that causes gastrointestinal symptoms [145], discriminating the edibility and poisoning of seven wild boletes mushrooms [146], and assessing the mortality rate of mushroom poisoning and its effects on the human liver [147].

## 8. Future Perspectives

There is a strong relationship between mushrooms (edible ones) and food, as many edible mushrooms are considered a source of healthy food and energy. To guide our discussion, we posed nine questions: (1) What is the relationship between mushrooms and human health? (2) What are the main factors controlling applications of mushrooms from farm to pharmacy? (3) What are the main factors controlling the edibility/toxicity of mushrooms? (4) To what extent can nanoparticles produced from mushroom contribute to the WEF nexus? (5) What are the promising roles of mushrooms in protecting the environment? (6) Are mushrooms a viable source of bioenergy production? (7) To what extent is the production of bioactives by mushrooms valuable? (8) Can mushrooms be integrated with crop production? (9) What new insights can be provided regarding mushrooms at the farming and pharma industry levels?

Most edible mushrooms belong to the Basidiomycetes, such as *Agaricus bisporus*, *Ganoderma lucidum*, *Flammulina velutipes*, *Lentinus edodes*, and *Pleurotus ostreatus* and *boletes*, while a few are ascomycetes, such as *Morchella esculenta*, *Cordyceps sinensis*, *Cordyceps militaris*, *Helvella elastica*, and truffles [115]. They can have a significant impact on human health due to their contents of bioactives and nutritional compounds. Certain species can have negative impacts up to and including death because of toxic compounds such as those found in poisonous mushrooms. The identification of mushrooms and their edibility is a critical subject [141].

The main factors controlling applications of mushrooms at the farm or pharmacy levels may include which mushroom species can be cultivated and which criteria are important to provide food, medicinal, or pharmaceutical benefits. Several mushrooms have been identified for edibility and many myco-chemicals have been identified in wild and cultivated mushrooms, with a focus on their nutritional and health benefits [148]. This may link to the criteria for the edibility/toxicity of mushrooms. This issue is still open and needs more investigation as not all species have been thoroughly studied, and their toxic or beneficial compounds are not yet fully understood [149]. Some recent studies reported on the controlling factors that make mushrooms edible and novel food products that contain mushrooms [150]. Some studies have started to highlight food poisoning from particular mushrooms and its mortality rate [147], whereas nephrotoxic poisoning by mushrooms and its global epidemiology was discussed by Diaz [151].

Research gaps concerning the WEF nexus and mushroom cultivation are presented in Figure 10. The nano-WEF nexus is a challenge, yet the application of nanotechnology in WEF resources can overcome this obstacle. The suggested role of myco-producing nanoparticles for the WEF nexus may open up a new window for these applications. Mushrooms have a role to play in achieving many SDGs, mainly linked to the goals based on the WEF nexus. In general, the biosynthesis of nanoparticles using mushrooms is a green and sustainable method which has already been applied for the remediation of polluted soil and water, but the application of nanofertilizers/nanopesticides for mushroom production still needs more investigation to produce healthy mushrooms as food. What is the promising role of mushrooms in protecting the environment? As mentioned before, mushrooms can produce NPs that are useful in environmental remediation (mainly soil and water), although the accumulation of SMS that results from the cultivation of mushrooms or mismanagement may lead to environmental problems. Regarding question six on mushrooms as a viable source for producing bioenergy, mushrooms have a potential role to play in producing bioenergy through myco-biorefinery by the bioconversion of food waste or mushroom waste to different value-added products [108].

Are the bioactives produced by mushrooms valuable? Yes, several bioactive compounds can be produced from mushroom extracts that benefit human health as emerging bioresources of nutraceuticals and food. These bioactive compounds have been successfully applied against many kinds of cancer and other human diseases. Regarding our question about the integrated production of mushrooms with cultivated crops, forests, or livestock, this is an important research area that is in need of more investigation. Such integration may support food and energy production as well as provide additional benefits (Table 6) [152].

Our last question is what are the new insights into mushrooms under/at the farming and pharma industry levels? As mentioned in the previous sections, mushrooms support many SDGs, mainly relating to food, energy, and water. Figure 11 presents a proposed integrated mushroom cultivation under the green circular agricultural system and its bioeconomy approach. This approach can be successfully applied to the sustainable bioeconomy. The model in Figure 11 presents several possible combinations between mushrooms (from one side) and the WEF nexus (from the other side). This model visualizes the interrelationships among mushrooms, food, and energy. This is an interrelationship that can control the production of food based on mushrooms or other sources of food and different possibilities for mushrooms in producing energy.

## 9. Conclusions

Mushrooms are a vital source of human food as part of a healthy and nutritious diet. Mushrooms do not need much water or energy during their cultivation compared to most crops. Thus, edible mushroom farming can contribute to achieving the UN’s sustainable development goals. Mushrooms are rich in bioactives and other pharmaceutical attributes. Mushrooms can increase food production (on the farm level) and contribute to the production of new medicines (on the pharmacy level). Mushrooms have a promising ability to remediate polluted soil and water. The crucial role of mushrooms in the water, energy, and food nexus has become increasingly clear; integrated mushroom farming may support global food security and guarantee more healthy food for many nations. Mushrooms are a particularly valuable resource that need more investigation to discover important qualities mainly in the fields of food, water, and energy for the future of humanity.

## Figures and Tables

**Figure 1 foods-12-02671-f001:**
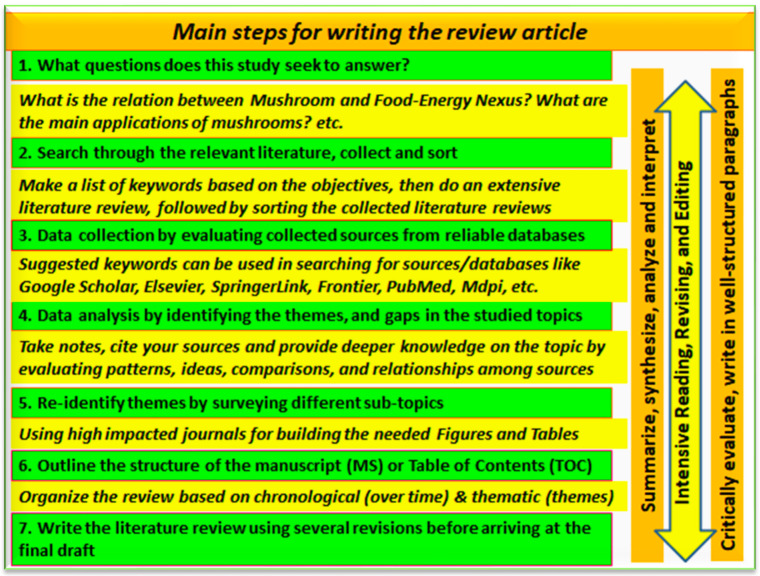
A flowchart showing how this study was conducted and the review prepared.

**Figure 2 foods-12-02671-f002:**
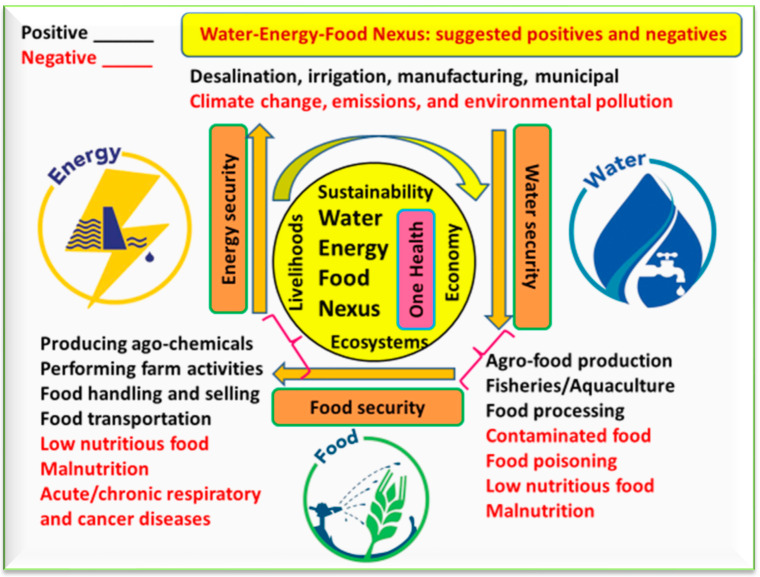
The concept of the water–energy–food nexus with focus on the positive (words in black) and negative (words in red) sides of the nexus on human and environmental health as society works towards sustainability.

**Figure 3 foods-12-02671-f003:**
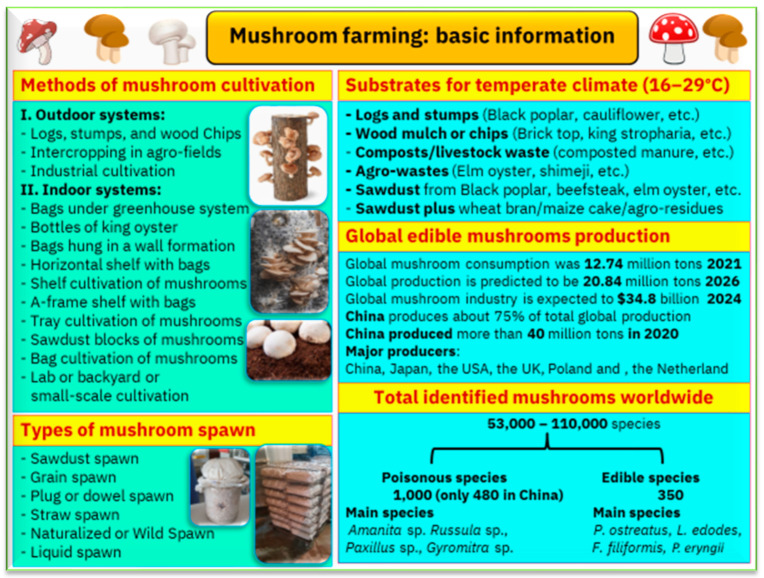
Basic information on mushrooms that include methods of cultivation, different kinds of substrates, different types of spawn, and alternative cultivation methods of mushrooms.

**Figure 4 foods-12-02671-f004:**
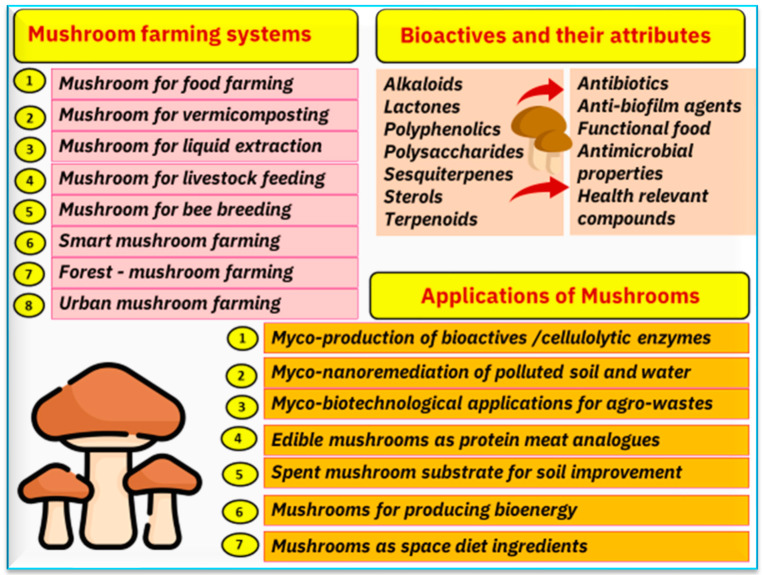
Mushrooms can be part of many farming systems, including food production, livestock feeding, forestry farming, and urban mushroom farming. Mushrooms have several applications such as the remediation of polluted soil and water, producing enzymes, bioactives (examples of them and possible attributes are given in the figure), and bioenergy.

**Figure 5 foods-12-02671-f005:**
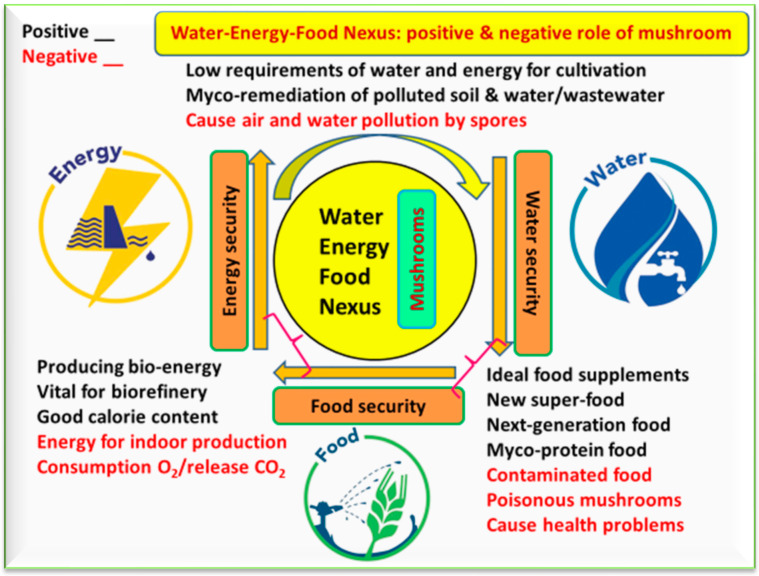
The relationships between mushrooms and the water–energy–food nexus, including positive and negative aspects. Mushrooms can support humanity with healthy food, but at the same time poisonous varieties can be toxic. They can produce bioenergy, but also consume O_2_ and release CO_2_ when cultivated indoors.

**Figure 6 foods-12-02671-f006:**
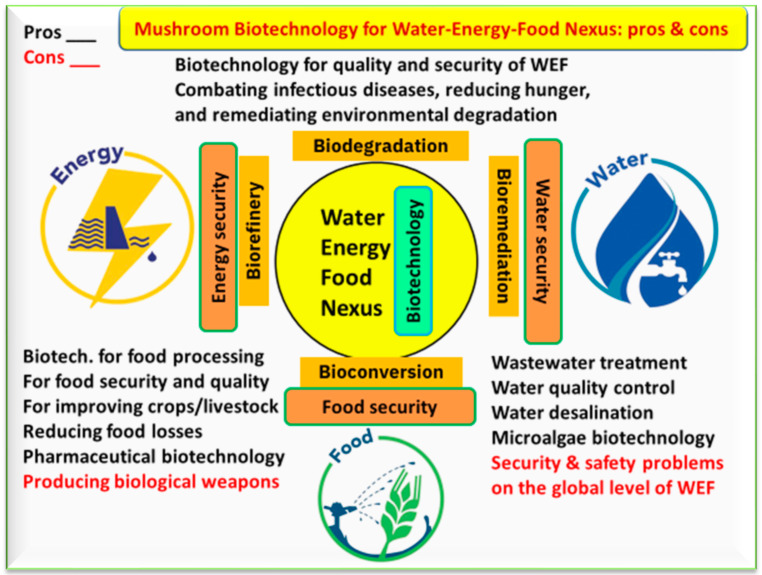
The role of mushroom biotechnology on the water–energy–food nexus, including the positive (pros in black color) and negative (cons in red color) aspects that may be generated due to different applications of biotechnology through certain processes, including bioconversion, biorefinery, bioremediation, and biodegradation.

**Figure 7 foods-12-02671-f007:**
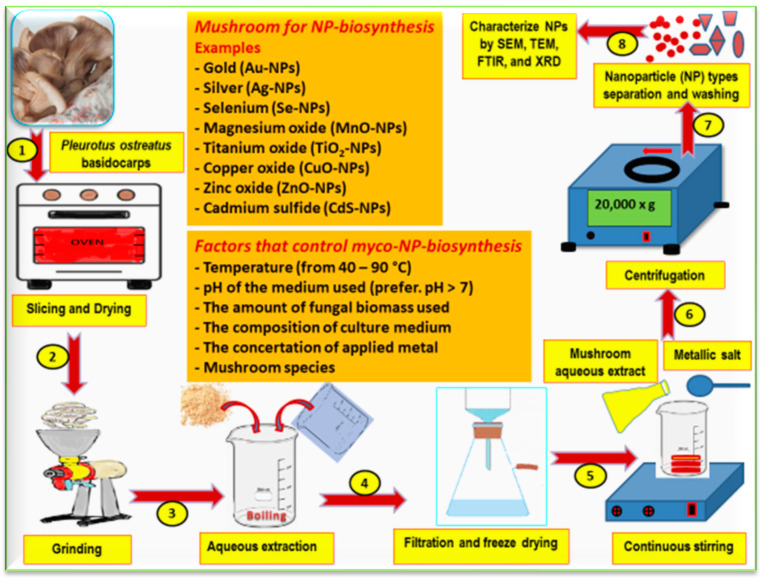
Steps for the biosynthesis of nanoparticles (NPs) from mushrooms (like oyster mushrooms) are as follows: (1) use of mushroom extracts as biomolecules or capping or reducing agents by slicing and then drying them in the oven (2), grinding these slices and using the mushroom powder extract to prepare the aqueous extraction (3); this extraction is then filtered and freeze dried (4). Mushroom extract was added into metal solution to reduce metallic ions from (M^+^) to (M^0^) via the oxidation/reduction mechanism with continuous stirring (5) and centrifugation (6) to form clusters of NPs that were confirmed after washing (7) and characterized using techniques (8) such as scanning electron microscopy (SEM), transmission electron microscopy (TEM), Fourier transform infrared spectroscopy (FTIR), and X-ray crystallography (XRD). Examples of NPs formed from mushrooms and factors that control this myco-formation are presented in the upper center of the figure (adapted from Elsakhawy et al.) [77].

**Figure 8 foods-12-02671-f008:**
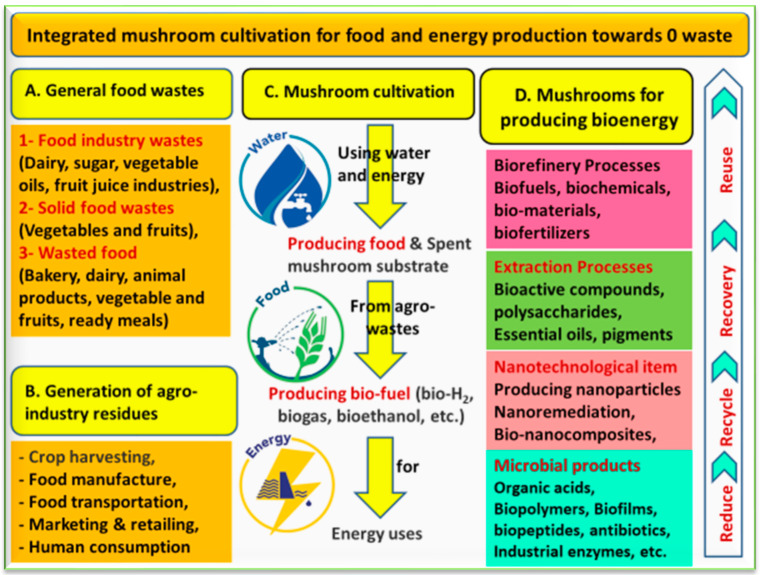
It is important to move towards zero (0) waste from food production and processing, and the mushroom industry can help with this. General food wastes are presented in (**A**), whereas the generation of agro-industry residues are in (**B**). The cultivation of mushrooms is an important source of healthy food (**C**). At the same time, producing only 1 kg of fresh mushroom may generate about 5 kg of wet byproducts, or spent mushroom substrate (SMS). Thus, this waste needs to be managed through the biorefinery or circular bioeconomic approach (**D**).

**Figure 9 foods-12-02671-f009:**
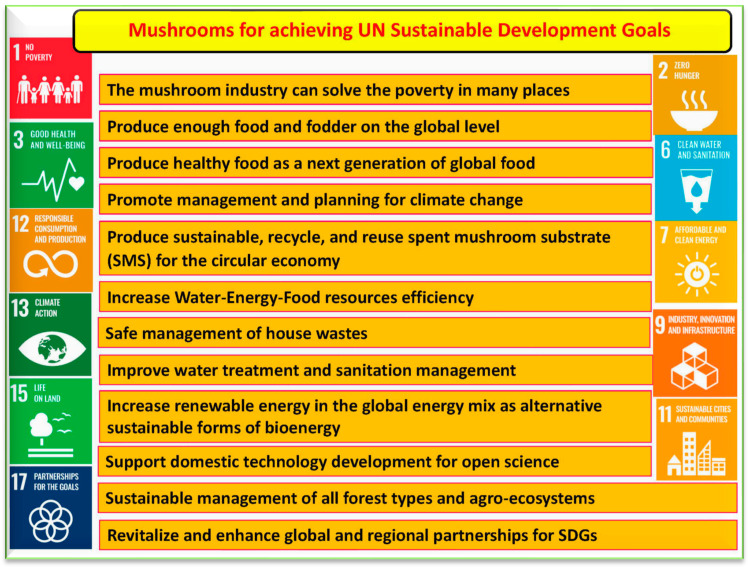
The crucial roles of mushrooms to help achieve many of the UN sustainable development goals, mainly the goals based on the water–energy–food nexus.

**Figure 10 foods-12-02671-f010:**
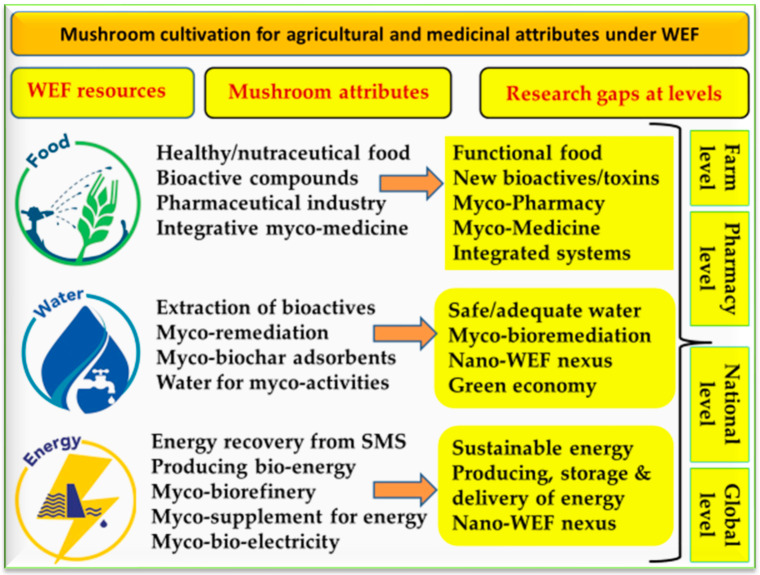
Summary of research gaps in the water–energy–food nexus related to mushroom cultivation, including different attributes of each WEF resource.

**Figure 11 foods-12-02671-f011:**
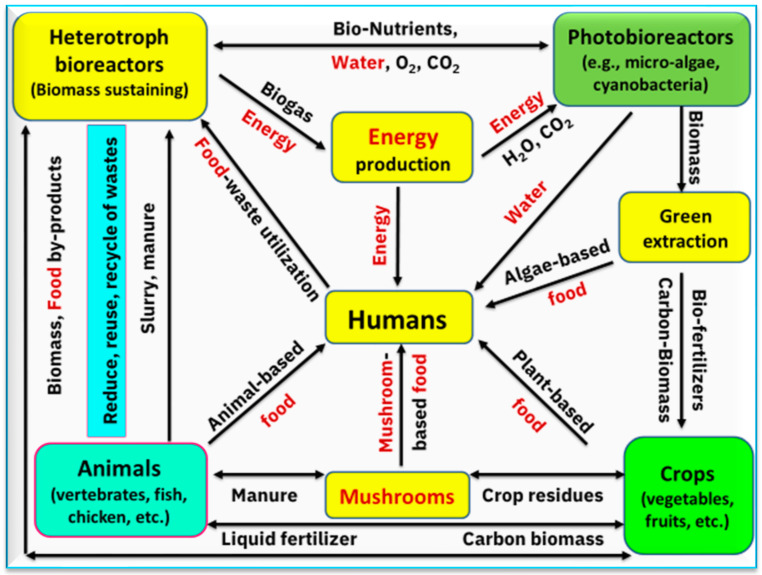
A suggested integrated mushroom cultivation plan in a green circular agricultural system and its bioeconomy approach for a sustainable bioeconomy model (adapted from Grimm et al. [160] and Sharma et al. [161]).

**Table 1 foods-12-02671-t001:** Some mushroom species and their health benefits as provided in recent references.

Mushroom Species	Health Benefits	References
Shiitake mushroom (*Lentinula edodes*)	Therapeutic potential that prevents diseases due to antiviral, antimicrobial, anticancer, antidiabetic, antiobesity, and antioxidant activities.	Ahmad et al. [17]
*Ganoderma* spp.	Have anti-inflammatory, antitumor, antioxidant, immune regulation, and other functions due to high polysaccharides content.	Yu et al. [57]
*Agaricus bisporus*	Protect against several human diseases due to many bioactive compounds (e.g., anticancer, anti-inflammation, and antioxidant agents).	Ahmed et al. [58]
Lion’s mane mushroom (*Hericium erinaceus*)	Antimicrobial action due to high content of corrinoids in the form of vitamin B_12_.	Rizzo et al. [59]
*Ganoderma applanatum*	Prevent oxidative stress (antioxidant) due to high ergothioneine content.	Martinez-Medina et al. [60]
Oyster mushroom (*Pleurotus ostreatus*)	Antimicrobial and prebiotic benefits for human health.	Xhensila et al. [61]
*Pleurotus ostreatus* Mushroom	Rich in chitin and glucan prebiotics, which enhance beneficial gut bacteria activity as antimicrobial agents.	Tör˝os et al. [62]
Morel mushroom (*Morchella esculenta* L.)	Health-promoting due to bioactives (polysaccharides and polynucleotides) that provide antidiabetic, antitumor, cardiovascular protective, antiparasitic, hepatoprotective, antibacterial, and antiviral properties.	Sunil and Xu [63]
Porcini mushroom (*Boletus edulis*)	Health-promoting effects from antineoplastic, antioxidant, antibacterial, anti-inflammatory, antiviral, and hepato-protective properties due to bioactive compounds.	Tan et al. [55]
*Russula griseocarnosa*	Health promoting actions include delaying aging and therapeutic functions (e.g., immune regulation, anticancer, antioxidant, hypoglycemic, and hypolipidemic activities).	Liu et al. [64]
*Tremella fuciformis*	Exopolysaccharides have may health benefits such as immune-enhancing, antitumor, and hypoglycemic properties.	Li et al. [65]
*Russula virescens*	Polysaccharides have potential anticancer, hypoglycemia, and immune-boosting activities by inhibiting α-glucosidase and α-amylase and mediating cellular immune response.	Li et al. [66]

**Table 3 foods-12-02671-t003:** Selected studies on mushroom remediation of polluted soil and water using spent mushroom substrate (SMS).

Mushroom Species	Polluted Medium	Main Finding of the Application	Reference
SMS from *Pleurotus eryngii* and *A. bisporus*	Cd-polluted paddy soil (total Cd, 72.87 mg kg^−1^)	Applied SMS improved the biomass of root and straw at different growth stages by reducing the uptake of Cd and accumulation in rice parts.	[99]
SMW of *Pleurotus ostreatus*	Anionic dyes with initial dose of 100–1300 mg g^−1^	Max. adsorption capacities of SMW were found to be 15.46, 18, 14.62, and 20.19 mg g^−1^ for DB22, DR5B, RB5, and DB71, respectively.	[100]
Spent mushroom substrate compost (SMSC) or biochar (SMSB)	Added 0.6, 1.2, 1.8, and 2.4 mg kg^−1^ Cd to soil	About 4% SMS can be used for amending Cd-polluted soils by Cd immobilization and improving chemical and biological soil properties.	[101]
SMS from *Pleurotus ostreatus*	Soil contained 8.535 SMS, and its applied rate was 20–40 mg kg^−1^	Optimum applied into the SMS is 8.86–9.51 g kg^−1^ soil when growing pak choi (*Brassica chinensis* L.).	[102]
SMS of *Pleurotus ostreatus*	Wastewater polluted with sulfonamides	Up to 83–91% of sulfonamides were removed over 14 days sulfamethoxazole, sulfathiazole, sulfadiazine, sulfapyridine, etc.	[103]
Spent mushroom substrate	Constructed wetland with simulated acid mine drainage	Removal rate of metal-burdened wastewater by SMS was Al, Zn, Cu (99%), Fe (97%), and Pb (97%) over a period of 800 days.	[104]
Spent mushroom substrate 0.5% (*w*/*w*)	Cd polluted soil, level at 0.6 mg kg^−1^	Applied SMS and biochar was more efficient than lime in reducing Cd content and increasing organic matter and enzyme activity after 4 weeks.	[105]
Spent mushroom substrate	Soil contaminated with carbendazim	SMS applied to fungicide-polluted soil reduced soil carbendazim residues and significantly increased the total-N, OM, and microbial biomass in the soil.	[106]
Substrates of Enoki,*A. bisporus*, and *Auricularia auricula* (AAR)	Soil polluted with chemical fertilizer	AAR recorded the highest level of soil nutrients among the 3 SMS replacements (mineral fertilizer by 25%); reduced heavy metals contamination.	[107]
Spent mushroom substrate and its biochar	Cd polluted soil, level at 0.6 mg kg^−1^	Applied SMS and its biochar alleviated the adverse effects of Cd and N and increased pH, CEC, and OM content in the soil.	[65]

**Table 4 foods-12-02671-t004:** Some edible mushroom species and their primary bioactive compounds content.

Mushroom Species	Main Groups of Bioactive Compounds				Refs.
	Phenolics	Polysaccharides	Proteins	Triterpenoids	
*Cordyceps aegerita*	Proto-catechuic acid	Fucogalactan	Ageritin	Bovistols A-C	Citores et al. [129]
*Boletus edulis*	Gallic acid	Polysaccharides (BEBP-1)	β-Trefoil lectin	Boledulins A-C	Luo et al. [130]
*Agaricus bisporus*	Gallocatechin	Heteropolysaccharide ABP	Protein type FIIb-1	Ergosterol	Liu et al. [131]
*Lactarius deliciosus*	Syringic acid, vanillic acid	Polysaccharide (LDG-M)	Laccase	Azulene-type sesquiterpen	Su et al. [132]
*Coprinus comatus*	Flavones and flavonols	Modified polysaccharide	Laccases	Terpenoids	Nowakowski et al. [133]
*Pleurotus ostreatus*	Caffeic acid and ferulic acid	Mycelium polysaccharides	Concanavalin A	Ergosterol	Fu et al. [134]
*Pleurotus cornucopiae*	Gallic acid	β-glucan	Oligopeptides	Ergostane-type sterols	Lee et al. [135]
*Macrolepiota procera*	Proto-catechuic acid	Polysaccharides	β-Trefoil lectin	Lanostane triterpenoids	Chen et al. [136]

Abbreviations: FIP (immunomodulatory proteins); PEPE (*Pleurotus eryngii* purified polysaccharides); PSK (polysaccharide K); PSP (polysaccharide peptide); RVP (*Russula virescens* polysaccharide); LDG m (*Lactarius deliciosus* polysaccharides); BEPF (crude polysaccharides isolated from *B. edulis*).

**Table 5 foods-12-02671-t005:** The most important poisonous mushrooms, their classification, and species for each category.

Poisonous Mushroom Group	Target Organ(s) or Symptoms	Mushroom Species	Toxic Dose *	Main Mushroom Toxins	Ref.
1. Cytotoxic mushrooms	Liver and kidneys	*Amanita bisporigera*	LD_50_ 0.4–0.8 mg kg^−1^	Amanitin (amatoxins, phallotoxins, and virotoxins)	[139]
2. Neurotoxic mushrooms	Neuroexcitatory effects	*Amanita*, *Clitocybe*, *Inocybe*, *Psilocybe*	400 mg/kg psilocin	Psilocybins, muscarines, and isoxazole	[140]
3. Myotoxic mushrooms	Symptoms of rhabdomyolysis	*Russula subnigricans* and *Tricholoma equestre*	LD_50_ 63.7–88.3 mg/kg	Russuphelins and cycloprop-2-ene carboxylic acid	[141]
4. Metabolic or endocrine mushrooms	Disulfiram-like symptoms	*Coprinus, Coprinopsis*, and *Ampulloclitocybe*	10–50 mg kg^−1^ gyromitrin	Trichothecene and gyromitrin	[137]
5. Gastrointestinal irritant mushroom	Gastrointestinal poisoning	*Agaricus, Entoloma*, *Gomphus, Hebeloma,* etc.	** Poisoning is rarely fatal	Specific toxins did not identify, but toxic phenolic compounds may *Agaricus* sp.	[142]
6. Miscellaneous adverse mushrooms	Hemolytic poisoning	Example of *Paxillus involutus* (Batsch) Fr.	Symptoms after 2–3 h take to death	The toxin is unknown at present	[143]

* Body weight in white mice. ** Vomiting is a hallmark of poisoning by gastrointestinal irritant mushrooms.

**Table 6 foods-12-02671-t006:** Some examples of integrated food and energy production from cultivated crops and mushrooms.

Mushroom Species	Plant Species	Food or Energy	The Main Purpose of Study	Reference
*Pleurotus ostreatus*	Crop residues (cassava, common bean, maize, banana)	Food and mushroom production	Cropping yield first and using crop residues for mushroom production, besides fodder and compost by farmers.	[153]
*Pleurotus sajor-caju*, *P. ostreatus*, and *Pleurotus eryngii*	Peels from the processing of fruits (mango, bananas, pineapple, avocado, orange, and watermelon	Mushroom production	Using fruit waste materials as a low-cost method to produce edible mushrooms as a source for health-promoting compounds such as antioxidants.	[154]
Oyster mushrooms (*Pleurotus ostreatus*)	Faba bean (*Vicia faba* L.) hulls	Combined mushroom and feed production	Faba bean hulls as a substrate for mushroom production led to higher protein levels and feed production.	[155]
Oyster mushrooms (*Pleurotus ostreatus*)	Banana leaf-midrib sticks	Mushroom production	Banana sticks were submerged in a liquid mycelium culture to produce spawn as a promising alternative industrial application.	[156]
*Pleurotus ostreatus* or *A. bisporus*	Tomato (*Solanum lycopersicum*)	Integrated tomato and mushroom production	Spent mushroom substrate was applied as a nutrient source to feed tomato seedlings in an integrated co-production system.	[157]
*Agaricus subrufescens* or *A. bisporus*	Lettuce, cucumber, and tomato	Integrated vegetables and mushroom cultivation	Spent mushrooms were used as compost combined with vermicompost, green waste compost, and fertigation with liquid digestate of food waste.	[158]
King oyster (*Pleurotus eryngii*)	Romaine lettuces (*Lactuca sativa* L.)	Food production	Sustainable agro-system using CO_2_ from mushrooms to cultivate lettuce in a continuous system.	[159]

## Data Availability

All data are presented in the MS.

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
