# Peer review of "Biotechnological Applications of Mushrooms under the Water-Energy-Food Nexus: Crucial Aspects and Prospects from Farm to Pharmacy"

_foods, 2023, doi:10.3390/foods12142671_

Round 1

Reviewer 1 Report

This manuscript provides an insight into the biotechnological applications of fungi in the water-energy-food nexus. It also discusses how fungi can be used to address the challenges of water, energy and food (WEF) scarcity to human survival. This includes in particular the use of fungi for the production of nutritious and protein-rich foods, biofuels and the use of their residues for water remediation. A key aspect discussed in the manuscript is the importance of considering the WEF nexus when exploring the integration of mushroom cultivation. The authors suggest that integrating mushroom cultivation with other agricultural practices can help improve resource efficiency and reduce waste. The manuscript also discusses some of the potential benefits and challenges associated with the use of mushrooms in biotechnology. For example, while there are many potential applications for fungi, there are also concerns about their safety and sustainability. Overall, this manuscript provides a comprehensive overview of the current state of research on fungal biotechnology and its potential applications in addressing challenges related to water, energy and food security. It highlights the opportunities and challenges associated with this emerging field and provides valuable insights for researchers, policy makers and practitioners.

Including but not limited to, the following issues requiring revision by the author;

1. Figure 1 shows the authors' thinking in writing the manuscript, but the quality of the image is so poor that it is not clear, please update.

2. Line 49, 4e18 J does not indicate 4e18 J, please standardize the writing.

3. Fig. 2, which has the same problem as Fig. 1, the resolution is too low to see clearly.

4. Well, the resolution of all figures should be adjusted to ensure that the reader can see them clearly.

5. The DOI in the reference is not necessary.

6. Line 356-379, outlines the production of mushrooms as biologically active compounds, please cite the following literature to make this statement more adequate.

1)        Diverse Metabolites and Pharmacological Effects from the Basidiomycetes Inonotus hispidus. 10.3390/antibiotics11081097

2)        Secondary Metabolites of Bird's Nest Fungi: Chemical Structures and Biological Activities. 10.1021/acs.jafc.3c00904

7. line 387-391, Fig. 10 and its figures are reproduced, please delete one of them.

8. Agaricus in 882 should be italic.

9. Agaricus bisporus appears several times in the manuscript, and the form written after the first should be A. bisporus.

It is OK.

This manuscript provides an insight into the biotechnological applications of fungi in the water-energy-food nexus. It also discusses how fungi can be used to address the challenges of water, energy and food (WEF) scarcity to human survival. This includes in particular the use of fungi for the production of nutritious and protein-rich foods, biofuels and the use of their residues for water remediation. A key aspect discussed in the manuscript is the importance of considering the WEF nexus when exploring the integration of mushroom cultivation. The authors suggest that integrating mushroom cultivation with other agricultural practices can help improve resource efficiency and reduce waste. The manuscript also discusses some of the potential benefits and challenges associated with the use of mushrooms in biotechnology. For example, while there are many potential applications for fungi, there are also concerns about their safety and sustainability. Overall, this manuscript provides a comprehensive overview of the current state of research on fungal biotechnology and its potential applications in addressing challenges related to water, energy and food security. It highlights the opportunities and challenges associated with this emerging field and provides valuable insights for researchers, policy makers and practitioners. In line with the scope of this journal, it is a review worthy of publication.

Including but not limited to, the following issues requiring revision by the author;

1. Figure 1 shows the authors' thinking in writing the manuscript, but the quality of the image is so poor that it is not clear, please update.

2. Line 49, 4e18 J does not indicate 4e18 J, please standardize the writing.

3. Fig. 2, which has the same problem as Fig. 1, the resolution is too low to see clearly.

4. Well, the resolution of all figures should be adjusted to ensure that the reader can see them clearly.

5. The DOI in the reference is not necessary.

6. Line 356-379, outlines the production of mushrooms as biologically active compounds, please cite the following literature to make this statement more adequate.

1)        Diverse Metabolites and Pharmacological Effects from the Basidiomycetes Inonotus hispidus. 10.3390/antibiotics11081097

2)        Secondary Metabolites of Bird's Nest Fungi: Chemical Structures and Biological Activities. 10.1021/acs.jafc.3c00904

7. line 387-391, Fig. 10 and its figures are reproduced, please delete one of them.

8. Agaricus in 882 should be italic.

9. Agaricus bisporus appears several times in the manuscript, and the form written after the first should be A. bisporus.

Author Response

Response to Reviewer 1

Dear Reviewer 1#

Many thanks for your time and your comments!

We followed your comments one by one, and made changes based your comments, hoping all these changes improved the MS to be ready for publication!

Comments and Suggestions for Authors

This manuscript provides an insight into the biotechnological applications of fungi in the water-energy-food nexus. It also discusses how fungi can be used to address the challenges of water, energy and food (WEF) scarcity to human survival. This includes in particular the use of fungi for the production of nutritious and protein-rich foods, biofuels and the use of their residues for water remediation. A key aspect discussed in the manuscript is the importance of considering the WEF nexus when exploring the integration of mushroom cultivation. The authors suggest that integrating mushroom cultivation with other agricultural practices can help improve resource efficiency and reduce waste. The manuscript also discusses some of the potential benefits and challenges associated with the use of mushrooms in biotechnology. For example, while there are many potential applications for fungi, there are also concerns about their safety and sustainability. Overall, this manuscript provides a comprehensive overview of the current state of research on fungal biotechnology and its potential applications in addressing challenges related to water, energy and food security. It highlights the opportunities and challenges associated with this emerging field and provides valuable insights for researchers, policy makers and practitioners.

Response: Many thanks for your time and your comments!

Many thanks for your great analysis for our MS, as well!

Including but not limited to, the following issues requiring revision by the author;

Response: Many thanks for your comments!

We worked on your comments, revised, improved and revised the MS, hoping all our corrections will meet your concern!

  1. Figure 1 shows the authors' thinking in writing the manuscript, but the quality of the image is so poor that it is not clear, please update.

Response: Many thanks for your comments!

More changes in the figure were done, and please I would like to draw your attention that many figures and tables might change after submission the MS to the website of the journal due to the difference between our office version and the journal one!

One important thing more, as we did in our previous publications in mdpi (around 40), after the accepting of the MS, we can send our original ppt including all figures to the mdpi journal, and they can put in the published MS a high-resolution version of each figure!

Hoping our point of view is clear, many thanks!

  1. Line 49, 4e18 J does not indicate 4e18 J, please standardize the writing.

Response: Many thanks for your comments!

The value was standardized, thanks!

  1. Fig. 2, which has the same problem as Fig. 1, the resolution is too low to see clearly.

Response: Many thanks for your comments!

More changes in the figure were done, as explained before thanks!

  1. Well, the resolution of all figures should be adjusted to ensure that the reader can see them clearly.

Response: Many thanks for your comments!

More changes in the figure were done, as explained before thanks!

  1. The DOI in the reference is not necessary.

Response: Many thanks for your comments!

Please let us disagree with you in this comment, because some very recent published articles have not any details concerning the pages number and volume, So, the DOI in this case is urgent, many thanks for your understanding!

Anyway, some of the DOI were removed, when the mentioned details were existed!

  1. Line 356-379, outlines the production of mushrooms as biologically active compounds, please cite the following literature to make this statement more adequate.

Response: Many thanks for your comments!

1)        Diverse Metabolites and Pharmacological Effects from the Basidiomycetes Inonotus hispidus. 10.3390/antibiotics11081097

Response: Many thanks for your comments!

The suggested ref. was added, thanks!

2)        Secondary Metabolites of Bird's Nest Fungi: Chemical Structures and Biological Activities. 10.1021/acs.jafc.3c00904

Response: Many thanks for your comments!

The suggested ref. was added, thanks!

  1. Wang, Z.-x.; Feng, X.-l.; Liu, C.; Gao, J.-m.; Qi, J. Diverse Metabolites and Pharmacological Effects from the Basidiomycetes Inonotus hispidus. Antibiotics 2022, 11, 1097. https://doi.org/10.3390/ antibiotics11081097
  2. Qi, J.; Gao, Y.Q.; Kang, S.J.; Liu, C.; Gao, J.M. Secondary Metabolites of Bird's Nest Fungi: Chemical Structures and Biological Activities. J Agric Food Chem., 2023, 71(17), 6513-6524. doi: 10.1021/acs.jafc.3c00904.

 line 387-391, Fig. 10 and its figures are reproduced, please delete one of them.

Response: Many thanks for your comments!

One of them deleted as you requested, thanks!

  1. Agaricus in 882 should be italic.

Response: Many thanks for your comments!

Done, thanks!

  1. Agaricus bisporus appears several times in the manuscript, and the form written after the first should be A. bisporus.

Response: Many thanks for your comments!

Done, thanks!

Comments on the Quality of English Language

It is OK.

Response: Many thanks for your comments!

Response: Many thanks for your time and your comments, again!

Hoping our corrections based on your advices improved our MS, thanks again!

Reviewer 2 Report

This paper is well-organized and written. However, there is no deeper analysis of the subject and the relation between Mushrooms- WEF. I recommend the addition of a more “scientific” part.  The conclusion should be extended in order to give particular directions related to WEF Nexus and further research in every area.

Line 49, change the number expression as the base of 10, instead e.

Author Response

Response to Reviewer 2

Dear Reviewer 2#

Many thanks for your time and your comments!

Comments and Suggestions for Authors

This paper is well-organized and written. However, there is no deeper analysis of the subject and the relation between Mushrooms- WEF.

Response: Many thanks for your comments!

Many thanks for your encouragements. You are right, this MS is well-organized and written!

About your comment:

“there is no deeper analysis of the subject and the relation between Mushrooms- WEF”

Due to the topic of our MS did not publish before, so our analysis was done based on the available data, hoping more publications and deeper analysis for this very important topic, however, more improvements in the revised MS were achieved, thanks!

Because the nexus of water, energy and food needs to be studied in intensive manner in the future for the global future, which will depend on the global security of food, water and energy

Hoping our opinion is clear and many thanks in advance for your understanding!

I recommend the addition of a more “scientific” part. 

Response: Many thanks for your comments!

We added 2 Tables to the revised MS including many scientific names of mushrooms, their health benefits, and their bioactive compounds

Hoping, this part will meet your concern!

The conclusion should be extended in order to give particular directions related to WEF Nexus and further research in every area.

Response: Many thanks for your comments!

More improvements to the conclusion section were added in the revised MS, thanks!

Line 49, change the number expression as the base of 10, instead e.

Response: Many thanks for your comments!

Done, thanks!

We followed your comments one by one, and made changes based your comments, hoping all these changes improved the MS to be ready for publication!

Hoping our improvements will meet your concern, thanks again!

Reviewer 3 Report

The submitted review manuscript entitled “Biotechnological Applications of Mushrooms under Water-Energy-Food Nexus: Crucial Aspects and Prospects from Farm to Pharmacy” describes the importance of mushrooms in various aspects that contribute to the sustainable development goals of the United Nations policy. This manuscript is recommended for publication in Foods Journal after minor revision. Please find the reviewer's comments below.

1.     All images are unclear. Please check the image resolution; embossing the image is unnecessary because it causes the image to appear blurry.

2.     It is unnecessary to describe how to write a review in Section 2. Section 2 and Figure 1 should be removed. The author can briefly clarify in the last paragraph of the introduction that all information was collected during the last five years from highly impacting journals or journals with a good reputation.

3.     Remove the quotation marks from "deal food supplements," "new superfood," "next-generation food of the future," and "myco-protein food" on lines 166-167.

4.     Add the reference to the content lines 176-179.

5.     Figure 6 should be changed into a table with information such as mushroom species, health benefits, and references.

6.     Figure 8, there's no need to include a comma in the text that already has a heading in front. Please check it.

7.     In section 6.4, the author should add a table of mushrooms that produces the bioactive compounds detailing the mushroom species, bioactive compounds produced, concentration, and references.

8.     There is no need to discuss questions and answers one by one in section 8. This topic should be eliminated and replaced with future perspectives. According to the title, the author ought to describe the future possibilities of mushrooms, from agriculture to medical reasons.

Minor editing of English language required

Author Response

Response to Reviewer 3

Dear Reviewer 3#

Many thanks for your time and your comments!

We followed your comments one by one, and made changes based your comments, hoping all these changes improved the MS to be ready for publication!

Comments and Suggestions for Authors

The submitted review manuscript entitled “Biotechnological Applications of Mushrooms under Water-Energy-Food Nexus: Crucial Aspects and Prospects from Farm to Pharmacy” describes the importance of mushrooms in various aspects that contribute to the sustainable development goals of the United Nations policy. This manuscript is recommended for publication in Foods Journal after minor revision.

Response: Many thanks for your comments!

Many thanks for your so kind words, which give us a great support, thanks again!

Please find the reviewer's comments below.

Response: Many thanks for your comments! With a great pleasure!

  1. All images are unclear. Please check the image resolution; embossing the image is unnecessary because it causes the image to appear blurry.

Response: Many thanks for your comments!

More changes in all figures were done, and please I would like to draw your attention that many figures and tables might change after submission the MS to the website of the journal due to the difference between our office version and the journal one!

One important thing more, as we did in our previous publications in mdpi (a round 40), after the accepting of the MS, we can send our original ppt including all figures to the mdpi journal, and they can put in the published MS a high-resolution version of each figure!

Hoping our point of view is clear, many thanks!

  1. It is unnecessary to describe how to write a review in Section 2. Section 2 and Figure 1 should be removed. The author can briefly clarify in the last paragraph of the introduction that all information was collected during the last five years from highly impacting journals or journals with a good reputation.

Response: Many thanks for your comments!

Your comments “all information was collected during the last five years from highly impacting journals or journals with a good reputation” was added to the end of the introduction section as you requested!

About the Section 2 and Figure 1, some reviewers asked on comments on this figure and this section, so, please accept our apology to be not able to remove them!

Please, let them stay, please! many thanks in advance for your understanding!

  1. Remove the quotation marks from "deal food supplements," "new superfood," "next-generation food of the future," and "myco-protein food" on lines 166-167.

Response: Many thanks for your comments!

Done, thanks!

  1. Add the reference to the content lines 176-179.

Response: Many thanks for your comments!

Done, thanks!

  1. Figure 6 should be changed into a table with information such as mushroom species, health benefits, and references.

Response: Many thanks for your comments!

The figure was changed into Table 1, based on your request, thanks!

  1. Figure 8, there's no need to include a comma in the text that already has a heading in front. Please check it.

Response: Many thanks for your comments!

Done, thanks!

  1. In section 6.4, the author should add a table of mushrooms that produces the bioactive compounds detailing the mushroom species, bioactive compounds produced, concentration, and references.

Response: Many thanks for your comments!

We added the requested Table but to be including the main groups of bioactive compounds of some selected edible mushroom, thanks!

About the concentration, we did not get the enough time to finish from the revised MS within the limited time!

  1. There is no need to discuss questions and answers one by one in section 8. This topic should be eliminated and replaced with future perspectives. According to the title, the author ought to describe the future possibilities of mushrooms, from agriculture to medical reasons.

Response: Many thanks for your comments!

The title of this section was changed into “future perspectives” based on your request!

About your comment on “the future possibilities of mushrooms from agriculture to medical reasons”,

Please, more changes were applied, and let us checked all questions:

(1) What is the relationship between mushroom and human health? (Medical side)

(2) What are the main factors controlling applications of mushroom from farm to pharmacy? (both medical and agricultural sides)

(3) What are the main factors controlling the edibility/poisoning of mushrooms? (both)

(4) To what extend can producing nanoparticles by mushroom crucial for WEF nexus? (this is according to the title, as you asked in your comment)

(5) What is the promising role of mushroom in protecting the environment? (including agro- and medical purposes)

(6) Is mushroom a vital source for producing bioenergy? (based on agro-wastes)

(7) To what extend is producing bioactives by mush-room a treasure? (medical point)

(8) Can mushroom be integrated with crop production? (agro-point)

(9) What are the new insights of mushrooms under the farming and pharma industry level? (both)

So, we are walking in the same way of your comment, that to focus on the title of this MS including the agro- and medical sides!

More changes were added to this section, hoping will be accepted by you!

Comments on the Quality of English Language

Minor editing of English language required

Many thanks for your great analysis for our MS, as well!

Really, your comments helped us to improve the revised MS

Hoping our improvement will meet your concern, thanks again!

Reviewer 4 Report

Biotechnological Applications of Mushrooms under Water-Energy-Food Nexus: Crucial Aspects and Prospects from Farm to Pharmacy

The authors have generated a review of how fungi are used and related it to the water-food-energy nexus. The figures are well done but there are several main issues with the manuscript.

1) First, a mushroom is the fruit body of a fungus (fungi). Therefore, it would be better for the authors to use the word fungi or fungus for applications that rely on the organism. I would suggest carefully deciding if the application was using the fruit body (mushroom) or if the application is using the fungus.

2) Most sections, if not all, need further refinement in the way information flows. There are sentences that appear to provide information but are awkwardly placed. 

Examples:  

Lines 58-63: The last sentence appears awkwardly placed.

Lines 90-92:  Should be reviewed and stated in a clearer manner

Lines 122-136: These sentences are awkward as written.

Lines 165-167: The sentence is a awkward as written.      

In general, the review does cite a lot of articles, but the review manuscript needs to have a better overall flow and read well.

The author needs to review the entire manuscript for sentence clarity (grammar).

Author Response

Response to Reviewer 4

Dear Reviewer 4#

Many thanks for your time and your comments!

Some copy and paste of our revised MS is attached, please follow the reports, thanks

We followed your comments one by one, and made changes based your comments, hoping all these changes improved the MS to be ready for publication!

Comments and Suggestions for Authors

Biotechnological Applications of Mushrooms under Water-Energy-Food Nexus: Crucial Aspects and Prospects from Farm to Pharmacy

The authors have generated a review of how fungi are used and related it to the water-food-energy nexus. The figures are well done but there are several main issues with the manuscript.

Response: Many thanks for your comments!

Many thanks for your so kind words, thanks!

1) First, a mushroom is the fruit body of a fungus (fungi). Therefore, it would be better for the authors to use the word fungi or fungus for applications that rely on the organism. I would suggest carefully deciding if the application was using the fruit body (mushroom) or if the application is using the fungus.

Response: Many thanks for your comments! We added a paragraph in the introduction where we now explain that mushroom is the fruit body of a fungus and that there is a high number of species in the Fungi kingdom but not all of them produce mushrooms.

We however kept the word mushrooms in the title and the text. We think this is wise as our MS is focusing on the fruiting bodies and their applications as food and using the SMS for other applications. We did not discuss the fungi but on a group that produces mushrooms NOT a general talk on fungi! Our MS is a specific review on mushrooms cultivation, their Cons, and Pros., as well as their biotechnological applications!

The word Fungi is mentioned in this MS a few times only. In addition to the general clarification in the introduction, we talk about fungi when we described the mushrooms as macro-fungi, role of mushroom as anti-fungal agents, and so on.

Hoping our overview is clear now, and many thanks for your understanding!

2) Most sections, if not all, need further refinement in the way information flows. There are sentences that appear to provide information but are awkwardly placed.

Response: Many thanks for your comments! We have carefully edited the entire MS to avoid such problems and the manuscript has thereafter been proofread.

Examples: 

Lines 58-63: The last sentence appears awkwardly placed.

Response: Many thanks for your comments! We now reformulated this section to:

Lines 90-92:  Should be reviewed and stated in a clearer manner

Response: Many thanks for your comments! We now say:

Lines 122-136: These sentences are awkward as written.

Response: Many thanks for your comments! We now say:

Lines 165-167: The sentence is a awkward as written.     

Response: Many thanks for your comments! We now say:

In general, the review does cite a lot of articles, but the review manuscript needs to have a better overall flow and read well.

Response: Many thanks for your comments!

We tried to follow, change, and improve the entire MS based on your advice, and we hope all these improvements will meet your concern! All changes are marked in the updated version.

Comments on the Quality of English Language

The author needs to review the entire manuscript for sentence clarity (grammar).

Response: The entire MS has been proofread and edited by a English speaking expert, and double checked again by us. This was done to improve the language and overall impression.

Many thanks for your great analysis for our MS, as well!

Really, your comments helped us to improve the revised MS

Hoping our improvement will meet your concern, thanks again!

Reviewer 5 Report

In the introduction, You should give more information about mushrooms and their application with a special aspect on the species of mushrooms that correlate with the topic of the paper.

Figure 1. is not needed in the paper.

All images should be adapted to the font suggested by the journal because they stand out too much in the paper.

Author Response

Response to Reviewer 5

Dear Reviewer 5#

Many thanks for your time and your comments!

Comments and Suggestions for Authors

In the introduction, You should give more information about mushrooms and their application with a special aspect on the species of mushrooms that correlate with the topic of the paper.

Response: Many thanks for your comments!

The requested part was added to the section of Introduction including the mushroom species that correlate with the topic of the paper.

Figure 1. is not needed in the paper.

Response: Many thanks for your comments!

About the figure 1, we totally respect your comment, but some reviewers have comments on this figure, if we delete it, this may cause a problem for the accepting our MS!!!

Please, we made some changes on this figure to be readable, please accept our apology, and many thanks for your understanding!

All images should be adapted to the font suggested by the journal because they stand out too much in the paper.

Response: Many thanks for your comments!

We have already used the Palatino Linotype as mentioned in instructions of the journal, but the added figure in the instructions of the journal did include any words and foe sure any font on the figure! We ask again the editor for the right instructions, and we will follow them, thanks!

One important thing more, as we did in our previous publications in mdpi (around 40), after the accepting of the MS, we can send our original ppt including all figures to the mdpi journal during the publication processes, and they can put in the published MS a high-resolution version of each figure and acceptable form by the journal! Hoping our point of view is clear, many thanks!

Really, your comments helped us to improve the revised MS

Hoping our improvement will meet your concern, thanks again!

Round 2

Reviewer 2 Report

Dear Authors,

When you reply to reviewers, don't use an "! "mark at the end of each comment, you seem unprofessional. You don't want to achieve that effect.

Best regards,